# Pruning Random Forests for Prediction on a Budget

**Feng Nan**
Systems Engineering
Boston University
fnan@bu.edu

**Joseph Wang**
Electrical Engineering
Boston University
joewang@bu.edu

**Venkatesh Saligrama**
Electrical Engineering
Boston University
srv@bu.edu

## Abstract

We propose to prune a random forest (RF) for resource-constrained prediction. We first construct a RF and then prune it to optimize expected feature cost & accuracy. We pose pruning RFs as a novel 0-1 integer program with linear constraints that encourages feature re-use. We establish total unimodularity of the constraint set to prove that the corresponding LP relaxation solves the original integer program. We then exploit connections to combinatorial optimization and develop an efficient primal-dual algorithm, scalable to large datasets. In contrast to our bottom-up approach, which benefits from good RF initialization, conventional methods are top-down acquiring features based on their utility value and is generally intractable, requiring heuristics. Empirically, our pruning algorithm outperforms existing state-of-the-art resource-constrained algorithms.

## 1 Introduction

Many modern classification systems, including internet applications (such as web-search engines, recommendation systems, and spam filtering) and security & surveillance applications (such as wide-area surveillance and classification on large video corpora), face the challenge of prediction-time budget constraints [21]. Prediction-time budgets can arise due to monetary costs associated with acquiring information or computation time (or delay) involved in extracting features and running the algorithm. We seek to learn a classifier by training on fully annotated training datasets that maintains high-accuracy while meeting average resource constraints during prediction-time. We consider a system that adaptively acquires features as needed depending on the instance(example) for high classification accuracy with reduced feature acquisition cost.

We propose a two-stage algorithm. In the first stage, we train a random forest (RF) of trees using an impurity function such as entropy or more specialized cost-adaptive impurity [16]. Our second stage takes a RF as input and attempts to jointly prune each tree in the forest to meet global resource constraints. During prediction-time, an example is routed through all the trees in the ensemble to the corresponding leaf nodes and the final prediction is based on a majority vote. The total feature cost for a test example is the sum of acquisition costs of *unique* features[1] acquired for the example in the entire ensemble of trees in the forest. [2]

We derive an efficient scheme to learn a *globally optimal pruning* of a RF minimizing the empirical error and incurred average costs. We formulate the pruning problem as a 0-1 integer linear program that incorporates feature-reuse constraints. By establishing total unimodularity of the constraint set, we show that solving the linear program relaxation of the integer program yields the optimal solution to the integer program resulting in a *polynomial*

*time algorithm for optimal pruning*. We develop a primal-dual algorithm by leveraging results from network-flow theory for scaling the linear program to large datasets. Empirically, this pruning outperforms state-of-the-art resource efficient algorithms on benchmarked datasets.

Our approach is motivated by the following considerations:

(**i**) RFs are scalable to large datasets and produce flexible decision boundaries yielding high prediction-time accuracy. The sequential feature usage of decision trees lends itself to adaptive feature acquisition. (**ii**) RF feature usage is superfluous, utilizing features with introduced randomness to increase diversity and generalization. Pruning can yield significant cost reduction with negligible performance loss by selectively pruning features sparsely used across trees, leading to cost reduction with minimal accuracy

|  | No Usage | 1–7 | > 7 | Cost | Error |
|---|---|---|---|---|---|
| Unpruned RF | 7.3% | 91.7% | 1% | 42.0 | 6.6% |
| BudgetPrune | 68.3% | 31.5% | 0.2% | 24.3 | 6.7% |

Table 1: Typical feature usage in a 40 tree RF before and after pruning (our algorithm) on the MiniBooNE dataset. Columns 2-4 list percentage of test examples that do not use the feature, use it 1 to 7 times, and use it greater than 7 times, respectively. Before pruning, 91% examples use the feature only a few (1 to 7) times, paying a significant cost for its acquisition; after pruning, 68% of the total examples no longer use this feature, reducing cost with minimal error increase. Column 5 is the average feature cost (the average number of unique features used by test examples). Column 6 is the test error of RFs. Overall, pruning dramatically reduces average feature cost while maintaining the same error level.

degradation (due to majority vote). See Table 1. (**iii**) Optimal pruning encourages examples to use features either a large number of times, allowing for complex decision boundaries in the space of those features, or not to use them at all, avoiding incurring the cost of acquisition. It enforces the fact that once a feature is acquired for an example, repeated use incurs no additional acquisition cost. Intuitively, features should be repeatedly used to increase discriminative ability without incurring further cost. (**iv**) Resource constrained prediction has been conventionally viewed as a top-down (tree-growing) approach, wherein new features are acquired based on their utility value. This is often an intractable problem with combinatorial (feature subsets) and continuous components (classifiers) requiring several relaxations and heuristics. In contrast, ours is a bottom-up approach that starts with good initialization (RF) and prunes to realize optimal cost-accuracy tradeoff. Indeed, while we do not pursue it, our approach can also be used in conjunction with existing approaches.

**Related Work:** Learning decision rules to minimize error subject to a budget constraint during prediction-time is an area of recent interest, with many approaches proposed to solve the prediction-time budget constrained problem [9, 22, 19, 20, 12]. These approaches focus on learning complex adaptive decision functions and can be viewed as orthogonal to our work. Conceptually, these are top-down "growing" methods as we described earlier (see (**iv**)). Our approach is bottom-up that seeks to prune complex classifiers to tradeoff cost vs. accuracy.

Our work is based on RF classifiers [3]. Traditionally, feature cost is not incorporated when constructing RFs, however recent work has involved approximation of budget constraints to learn budgeted RFs [16]. The tree-growing algorithm in [16] does not take feature re-use into account. Rather than attempting to approximate the budget constraint during tree construction, our work focuses on pruning ensembles of trees subject to a budget constraint. Methods such as traditional ensemble learning and budgeted random forests can be viewed as complementary.

Decision tree pruning has been studied extensively to improve generalization performance, we are not aware of any existing pruning method that takes into account the feature costs. A popular method for pruning to reduce generalization error is Cost-Complexity Pruning (CCP), introduced by Breiman et al. [4]. CCP trades-off classification ability for tree size, however it does not account for feature costs. As pointed out by Li et al. [15], CCP has undesirable "jumps" in the sequence of pruned tree sizes. To alleviate this, they proposed a Dynamic-Program-based Pruning (DPP) method for binary trees. The DPP algorithm is able to obtain optimally pruned trees of all sizes; however, it faces the curse of dimensionality when pruning an ensemble of decision trees and taking feature cost into account. [23, 18] proposed to solve the pruning problem as a 0-1 integer program; again, their formulations do not account for feature costs that we focus on in this paper. The coupling nature of feature usage makes our problem much harder. In general pruning RFs is not a focus of attention as it is assumed that overfitting can be avoided by constructing an ensemble of trees. While this is true, it often leads to extremely large prediction-time costs. Kulkarni and Sinha [11] provide a survey of methods to prune RFs in order to reduce ensemble size. However, these methods do not explicitly account for feature costs.

## 2 Learning with Resource Constraints

In this paper, we consider solving the Lagrangian relaxed problem of learning under prediction-time resource constraints, also known as the error-cost tradeoff problem:

$$\min_{f \in \mathcal{F}} E_{(x,y) \sim \mathcal{P}} \left[ err\left(y, f(x)\right) \right] + \lambda E_{x \sim \mathcal{P}_x} \left[ C\left(f, x\right) \right], \tag{1}$$

where example/label pairs $(x, y)$ are drawn from a distribution $\mathcal{P}$; $err(y, \hat{y})$ is the error function; $C(f, x)$ is the cost of evaluating the classifier $f$ on example $x$; $\lambda$ is a tradeoff parameter. A larger $\lambda$ places a larger penalty on cost, pushing the classifier to have smaller cost. By adjusting $\lambda$ we can obtain a classifier satisfying the budget constraint. The family of classifiers $\mathcal{F}$ in our setting is the space of RFs, and each RF $f$ is composed of $T$ decision trees $\mathcal{T}_1, \ldots, \mathcal{T}_T$.

**Our approach:** Rather than attempting to construct the optimal ensemble by solving Eqn. (1) directly, we instead propose a two-step algorithm that first constructs an ensemble with low prediction error, then prunes it by solving Eqn. (1) to produce a pruned ensemble given the input ensemble. By adopting this two-step strategy, we obtain an ensemble with low expected cost while simultaneously preserving the low prediction error.

There are many existing methods to construct RFs, however the focus of this paper is on the second step, where we propose a novel approach to prune RFs to solve the tradeoff problem Eqn.(1). Our pruning algorithm is capable of taking any RF as input, offering the flexibility to incorporate any state-of-the-art RF algorithm.

## 3 Pruning with Costs

In this section, we treat the error-cost tradeoff problem Eqn. (1) as an RF pruning problem. Our key contribution is to formulate pruning as a 0-1 integer program with totally unimodular constraints.

We first define notations used throughout the paper. A training sample $S = \{(\mathbf{x}^{(i)}, y^{(i)}) : i = 1, \ldots, N\}$ is generated i.i.d. from an unknown distribution, where $\mathbf{x}^{(i)} \in \Re^K$ is the feature vector with a cost assigned to each of the $K$ features and $y^{(i)}$ is the label for the $i$th example. In the case of multi-class classification $y \in \{1, \ldots, M\}$, where $M$ is the number of classes. Given a decision tree $\mathcal{T}$, we index the nodes as $h \in \{1, \ldots, |\mathcal{T}|\}$, where node 1 represents the root node. Let $\tilde{\mathcal{T}}$ denote the set of leaf nodes of tree $\mathcal{T}$. Finally, the corresponding definitions for $\mathcal{T}$ can be extended to an ensemble of $T$ decision trees $\{\mathcal{T}_t : t = 1, \ldots, T\}$ by adding a subscript $t$.

**Pruning Parametrization:** In order to model ensemble pruning as an optimization problem, we parametrize the space of all prunings of an ensemble. The process of pruning a decision tree $\mathcal{T}$ at an internal node $h$ involves collapsing the subtree of $\mathcal{T}$ rooted at $h$, making $h$ a leaf node. We say a pruned tree $\mathcal{T}^{(p)}$ is a valid pruned tree of $\mathcal{T}$ if (1) $\mathcal{T}^{(p)}$ is a subtree of $\mathcal{T}$ containing root node 1 and (2) for any $h \neq 1$ contained in $\mathcal{T}^{(p)}$, the sibling nodes (the set of nodes that share the same immediate parent node as $h$ in $\mathcal{T}$) must also be contained in $\mathcal{T}^{(p)}$. Specifying a pruning is equivalent to specifying the nodes that are leaves in the pruned tree. We therefore introduce the following binary variable for each node $h \in \mathcal{T}$

$$z_h = \begin{cases} 1 & \text{if node } h \text{ is a leaf in the pruned tree,} \\ 0 & \text{otherwise.} \end{cases}$$

We call the set $\{z_h, \forall h \in \mathcal{T}\}$ the node variables as they are associated with each node in the tree. Consider any root-to-leaf path in a tree $\mathcal{T}$, there should be exactly one node in the path that is a leaf node in the pruned tree. Let $p(h)$ denote the set of predecessor nodes, the set of nodes (including $h$) that lie on the path from the root node to $h$. The set of valid pruned trees can be represented as the set of node variables satisfying the following set of constraints: $\sum_{u \in p(h)} z_u = 1 \quad \forall h \in \tilde{\mathcal{T}}$. Given a valid pruning for a tree, we now seek to parameterize the error of the pruning.

**Pruning error:** As in most supervised empirical risk minimization problems, we aim to minimize the error on training data as a surrogate to minimizing the expected error. In a decision tree $\mathcal{T}$, each node $h$ is associated with a predicted label corresponding to the majority label among the training examples that fall into the node $h$. Let $S_h$ denote the subset of examples in $S$ routed to or through node $h$ on $\mathcal{T}$ and let $\text{Pred}_h$ denote the predicted label at $h$. The number of misclassified examples

at $h$ is therefore $e_h = \sum_{i \in S_h} \mathbb{1}_{\left[y^{(i)} \neq \text{Pred}_h\right]}$. We can thus estimate the error of tree $\mathcal{T}$ in terms of the number of misclassified examples in the leaf nodes: $\frac{1}{N} \sum_{h \in \tilde{\mathcal{T}}} e_h$, where $N = |S|$ is the total number of examples.

Our goal is to minimize the expected test error of the trees in the random forest, which we empirically approximate based on the aggregated probability distribution in Step (6) of Algorithm 1 with $\frac{1}{TN} \sum_{t=1}^{T} \sum_{h \in \tilde{\mathcal{T}}_t} e_h$. We can express this error in terms of the node variables: $\frac{1}{TN} \sum_{t=1}^{T} \sum_{h \in \mathcal{T}_t} e_h z_h$.

**Pruning cost:** Assume the acquisition costs for the $K$ features, $\{c_k : k = 1, \ldots, K\}$, are given. The feature acquisition cost incurred by an example is the sum of the acquisition costs of unique features acquired in the process of running the example through the forest. This cost structure arises due to the assumption that an acquired feature is cached and subsequent usage by the same example incurs no additional cost. Formally, the feature cost of classifying an example $i$ on the ensemble $\mathcal{T}_{[T]}$ is given by $C_{\text{feature}}(\mathcal{T}_{[T]}, \mathbf{x}^{(i)}) = \sum_{k=1}^{K} c_k w_{k,i}$, where the binary variables $w_{k,i}$ serve as the indicators:

$$w_{k,i} = \begin{cases} 1 & \text{if feature } k \text{ is used by } \mathbf{x}^{(i)} \text{ in any } \mathcal{T}_t, t = 1, \ldots, T \\ 0 & \text{otherwise.} \end{cases}$$

The expected feature cost of a test example can be approximated as $\frac{1}{N} \sum_{i=1}^{N} \sum_{k=1}^{K} c_k w_{k,i}$.

In some scenarios, it is useful to account for computation cost along with feature acquisition cost during prediction-time. In an ensemble, this corresponds to the expected number of Boolean operations required running a test through the trees, which is equal to the expected depth of the trees. This can be modeled as $\frac{1}{N} \sum_{t=1}^{T} \sum_{h \in \mathcal{T}_t} |S_h| d_h z_h$, where $d_h$ is the depth of node $h$.

**Putting it together:** Having modeled the pruning constraints, prediction performance and costs, we formulate the problem of pruning using the relationship between the node variables $z_h$'s and feature usage variables $w_{k,i}$'s. Given a tree $\mathcal{T}$, feature $k$, and example $\mathbf{x}^{(i)}$, let $u_{k,i}$ be the first node associated with feature $k$ on the root-to-leaf path the example follows in $\mathcal{T}$. Feature $k$ is used by $\mathbf{x}^{(i)}$ if and only if none of the nodes between the root and $u_{k,i}$ is a leaf. We represent this by the constraint $w_{k,i} + \sum_{h \in p(u_{k,i})} z_h = 1$ for every feature $k$ used by example $x^{(i)}$ in $\mathcal{T}$. Recall $w_{k,i}$ indicates whether or not feature $k$ is used by example $i$ and $p(u_{k,i})$ denotes the set of predecessor nodes of $u_{k,i}$. Intuitively, this constraint says that either the tree is pruned along the path followed by example $i$ before feature $k$ is acquired, in which case $z_h = 1$ for some node $h \in p(u_{k,i})$ and $w_{k,i} = 0$; or $w_{k,i} = 1$, indicating that feature $k$ is acquired for example $i$. We extend the notations to ensemble pruning with tree index $t$: $z_h^{(t)}$ indicates whether node $h$ in $\mathcal{T}_t$ is a leaf after pruning; $w_{k,i}^{(t)}$ indicates whether feature $k$ is used by the $i^{\text{th}}$ example in $\mathcal{T}_t$; $w_{k,i}$ indicates whether feature $k$ is used by the $i^{\text{th}}$ example in any of the $T$ trees $\mathcal{T}_1, \ldots, \mathcal{T}_T$; $u_{t,k,i}$ is the first node associated with feature $k$ on the root-to-leaf path the example follows in $\mathcal{T}_t$; $K_{t,i}$ denotes the set of features the $i^{\text{th}}$ example uses on tree $\mathcal{T}_t$. We arrive at the following integer program.

$$\min_{\substack{z_h^{(t)}, w_{k,i}^{(t)}, \\ w_{k,i} \in \{0,1\}}} \overbrace{\frac{1}{NT} \sum_{t=1}^{T} \sum_{h \in \mathcal{T}_t} e_h^{(t)} z_h^{(t)}}^{\text{error}} + \lambda \left( \overbrace{\sum_{k=1}^{K} c_k \left( \frac{1}{N} \sum_{i=1}^{N} w_{k,i} \right)}^{\text{feature acquisition cost}} + \overbrace{\frac{1}{N} \sum_{t=1}^{T} \sum_{h \in \mathcal{T}_t} |S_h| d_h z_h}^{\text{computational cost}} \right) \quad \textbf{(IP)}$$

$$\text{s.t.} \quad \sum_{u \in p(h)} z_u^{(t)} = 1, \qquad \forall h \in \tilde{\mathcal{T}}_t, \forall t \in [T], \qquad \text{(feasible prunings)}$$

$$w_{k,i}^{(t)} + \sum_{h \in p(u_{t,k,i})} z_h^{(t)} = 1, \quad \forall k \in K_{t,i}, \forall i \in S, \forall t \in [T], \text{ (feature usage/ tree)}$$

$$w_{k,i}^{(t)} \leq w_{k,i}, \qquad \forall k \in [K], \forall i \in S, \forall t \in [T]. \text{ (global feature usage)}$$

**Totally Unimodular constraints:** Even though integer programs are NP-hard to solve in general, we show that **(IP)** can be solved exactly by solving its LP relaxation. We prove this in two steps: first, we examine the special structure of the equality constraints; then we examine the inequality constraint that couples the trees. Recall that a network matrix is one with each column having exactly one element equal to 1, one element equal to -1 and the remaining elements being 0. A network matrix defines a directed graph with the nodes in the rows and arcs in the columns. We have the following lemma.

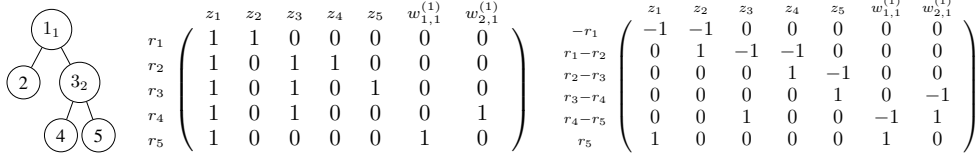

Figure 1: A decision tree example with node numbers and associated feature in subscripts together with the constraint matrix and its equivalent network matrix form.

**Lemma 3.1** *The equality constraints in* **(IP)** *can be turned into an equivalent network matrix form for each tree.*

**Proof** We observe the first constraint $\sum_{u \in p(h)} z_u^{(t)} = 1$ requires the sum of the node variables along a path to be 1. The second constraints $w_{k,i}^{(t)} + \sum_{h \in p(u_{t,k,i})} z_h^{(t)} = 1$ has a similar sum except the variable $w_{k,i}^{(t)}$. Imagine $w_{k,i}^{(t)}$ as yet another node variable for a fictitious child node of $u_{t,k,i}$ and the two equations are essentially equivalent. The rest of proof follows directly from the construction in Proposition 3 of [18].

Figure 1 illustrates such a construction. The nodes are numbered 1 to 5. The subscripts at node 1 and 3 are the feature index used in the nodes. Since the equality constraints in **(IP)** can be separated based on the trees, we consider only one tree and one example being routed to node 4 on the tree for simplicity. The equality constraints can be organized in the matrix form as shown in the middle of Figure 1. Through row operations, the constraint matrix can be transformed to an equivalent network matrix. Such transformation always works as long as the leaf nodes are arranged in a pre-order manner. Next, we deal with the inequality constraints and obtain our main result.

**Theorem 3.2** *The LP relaxation of* **(IP)***, where the 0-1 integer constraints are relaxed to interval constraints* $[0, 1]$ *for all integer variables, has integral optimal solutions.*

Due to space limit the proof can be found in the Suppl. Material. The main idea is to show the constraints are still *totally unimodular* even after adding the coupling constraints and the LP relaxed polyhedron has only integral extreme points [17]. As a result, solving the LP relaxation results in the optimal solution to the integer program **(IP)**, allowing for polynomial time optimization. [3]

---

**Algorithm 1 BUDGETPRUNE**

---

**During Training: input - ensemble($\mathcal{T}_1, \ldots, \mathcal{T}_T$), training/validation data with labels, $\lambda$**

---

1: initialize dual variables $\beta_{k,i}^{(t)} \leftarrow 0$.
2: update $z_h^{(t)}, w_{k,i}^{(t)}$ for each tree $t$ (shortest-path algo). $w_{k,i} = 0$ if $\mu_{k,i} > 0$, $w_{k,i} = 1$ if $\mu_{k,i} < 0$.
3: $\beta_{k,i}^{(t)} \leftarrow [\beta_{k,i}^{(t)} + \gamma(w_{k,i}^{(t)} - w_{k,i})]_+$ for step size $\gamma$, where $[\cdot]_+ = \max\{0, \cdot\}$.
4: go to Step 2 until duality gap is small enough.

---

**During Prediction: input - test example x**

---

5: Run **x** on each tree to leaf, obtain the probability distribution over label classes $\mathbf{p}_t$ at leaf.
6: Aggregate $\mathbf{p} = \frac{1}{T}\sum_{t=1}^{T} \mathbf{p}_t$. Predict the class with the highest probability in $\mathbf{p}$.

---

## 4 A Primal-Dual Algorithm

Even though we can solve **(IP)** via its LP relaxation, the resulting LP can be too large in practical applications for any general-purpose LP solver. In particular, the number of variables and constraints is roughly $O(T \times |\mathcal{T}_{\max}| + N \times T \times K_{\max})$, where $T$ is the number of trees; $|\mathcal{T}_{\max}|$ is the maximum

number of nodes in a tree; $N$ is the number of examples; $K_{\max}$ is the maximum number of features an example uses in a tree. The runtime of the LP thus scales $O(T^3)$ with the number of trees in the ensemble, limiting the application to only small ensembles. In this section we propose a primal-dual approach that effectively decomposes the optimization into many sub-problems. Each sub-problem corresponds to a tree in the ensemble and can be solved efficiently as a shortest path problem. The runtime per iteration is $O(\frac{T}{p}(|\mathcal{T}_{\max}| + N \times K_{\max})\log(|\mathcal{T}_{\max}| + N \times K_{\max}))$, where $p$ is the number of processors. We can thus massively parallelize the optimization and scale to much larger ensembles as the runtime depends only linearly on $\frac{T}{p}$. To this end, we assign dual variables $\beta_{k,i}^{(t)}$ for the inequality constraints $w_{k,i}^{(t)} \leq w_{k,i}$ and derive the dual problem.

$$\max_{\substack{\beta_{k,i}^{(t)} \geq 0}} \min_{\substack{z_h^{(t)} \in [0,1] \\ w_{k,i}^{(t)} \in [0,1] \\ w_{k,i} \in [0,1]}} \quad \frac{1}{NT} \sum_{t=1}^{T} \sum_{h \in \mathcal{T}_t} \hat{e}_h^{(t)} z_h^{(t)} + \lambda \left( \sum_{k=1}^{K} c_k \left( \frac{1}{N} \sum_{i=1}^{N} w_{k,i} \right) \right) + \sum_{t=1}^{T} \sum_{i=1}^{N} \sum_{k \in K_{t,i}} \beta_{k,i}^{(t)} (w_{k,i}^{(t)} - w_{k,i})$$

$$\text{s.t.} \quad \sum_{u \in p(h)} z_u^{(t)} = 1, \qquad \forall h \in \tilde{\mathcal{T}}_t, \forall t \in [T],$$

$$w_{k,i}^{(t)} + \sum_{h \in p(u_{t,k,i})} z_h^{(t)} = 1, \quad \forall k \in K_{t,i}, \forall i \in S, \forall t \in [T],$$

where for simplicity we have combined coefficients of $z_h^{(t)}$ in the objective of (**IP**) to $\hat{e}_h^{(t)}$. The primal-dual algorithm is summarized in Algorithm 1. It alternates between updating the primal and the dual variables. The key is to observe that given dual variables, the primal problem (inner minimization) can be decomposed for each tree in the ensemble and solved in parallel as shortest path problems due to Lemma 3.1. (See also Suppl. Material). The primal variables $w_{k,i}$ can be solved in closed form: simply compute $\mu_{k,i} = \lambda c_k/N - \sum_{t \in T_{k,i}} \beta_{k,i}^{(t)}$, where $T_{k,i}$ is the set of trees in which example $i$ encounters feature $k$. So $w_{k,i}$ should be set to 0 if $\mu_{k,i} > 0$ and $w_{k,i} = 1$ if $\mu_{k,i} < 0$.

Note that our prediction rule aggregates the leaf distributions from all trees instead of just their predicted labels. In the case where the leaves are pure (each leaf contains only one class of examples), this prediction rule coincides with the majority vote rule commonly used in random forests. Whenever the leaves contain mixed classes, this rule takes into account the prediction confidence of each tree in contrast to majority voting. Empirically, this rule consistently gives lower prediction error than majority voting with pruned trees.

## 5 Experiments

We test our pruning algorithm BUDGETPRUNE on four benchmark datasets used for prediction-time budget algorithms. The first two datasets have unknown feature acquisition costs so we assign costs to be 1 for all features; the aim is to show that BUDGETPRUNE successfully selects a sparse subset of features on average to classify each example with high accuracy. [4] The last two datasets have real feature acquisition costs measured in terms of CPU time. BUDGETPRUNE achieves high prediction accuracy spending much less CPU time in feature acquisition.

For each dataset we first train a RF and apply BUDGETPRUNE on it using different $\lambda$'s to obtain various points on the accuracy-cost tradeoff curve. We use in-bag data to estimate error probability at each node and the validation data for the feature cost variables $w_{k,i}$'s. We implement BUDGETPRUNE using CPLEX [1] network flow solver for the primal update step. The running time is significantly reduced (from hours down to minutes) compared to directly solving the LP relaxation of (**IP**) using standard solvers such as Gurobi [10]. Futhermore, the standard solvers simply break trying to solve the larger experiments whereas BUDGETPRUNE handles them with ease. We run the experiments for 10 times and report the means and standard deviations. Details of datasets and parameter settings of competing methods are included in the Suppl. Material.

**Competing methods:** We compare against four other approaches. (**i**) BUDGETRF[16]: the recursive node splitting process for each tree is stopped as soon as node impu-

rity (entropy or Pairs) falls below a threshold. The threshold is a measure of impurity tolerated in the leaf nodes. This can be considered as a naive pruning method as it reduces feature acquisition cost while maintaining low impurity in the leaves.

(**ii**) Cost-Complexity Pruning (CCP) [4]: it iteratively prunes subtrees such that the resulting tree has low error and small size. We perform CCP on individual trees to different levels to obtain various points on the accuracy-cost tradeoff curve. CCP does not take into account feature costs. (**iii**) GREEDYPRUNE: is a greedy global feature pruning strategy that we propose; at each iteration it attempts to remove all nodes corresponding to one feature from the RF such that the resulting

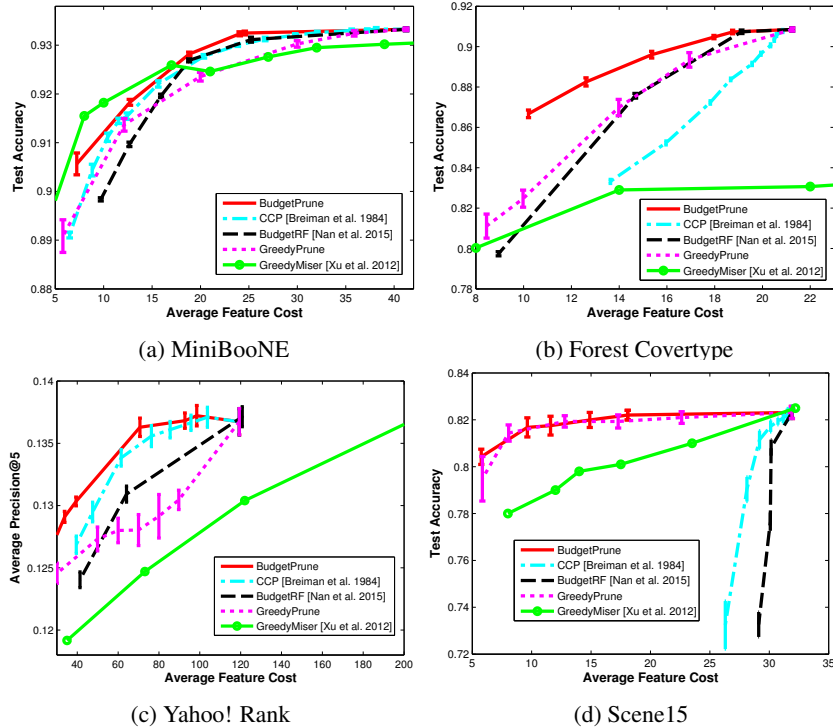

(a) MiniBooNE

(b) Forest Covertype

(c) Yahoo! Rank

(d) Scene15

Figure 2: Comparison of BUDGETPRUNE against CCP, BUDGETRF with early stopping, GREEDYPRUNE and GREEDYMISER on 4 real world datasets. BUDGETPRUNE (red) outperforms competing state-of-art methods. GREEDYMISER dominates ASTC [12], CSTC [21] and DAG [20] significantly on all datasets. We omit them in the plots to clearly depict the differences between competing methods.

pruned RF has the lowest training error and average feature cost. The process terminates in at most K iterations, where K is the number of features. The idea is to reduce feature costs by successively removing features that result in large cost reduction yet small accuracy loss. We also compare against the state-of-the-art methods in budgeted learning (**iv**) GREEDYMISER [22]: it is a modification of gradient boosted regression tree [8] to incorporate feature cost. Specifically, each weak learner (a low-depth decision tree) is built to minimize squared loss with respect to current gradient at the training examples plus feature acquisition cost. To build each weak learner the feature costs are set to zero for those features already used in previous weak learners. Other prediction-time budget algorithms such as ASTC [12], CSTC [21] and cost-weighted $l$-1 classifiers are shown to perform strictly worse than GREEDYMISER by a significant amount [12, 16] so we omit them in our plots. Since only the feature acquisition costs are standardized, for fair comparison we do not include the computation cost term in the objective of (**IP**) and focus instead on feature acquisition costs.

**MiniBooNE Particle Identification and Forest Covertype Datasets:[7]** Feature costs are uniform in both datasets. Our base RF consists of 40 trees using entropy split criteria and choosing from the full set of features at each split. As shown in (a) and (b) of Figure 2, BUDGETPRUNE (in red) achieves the best accuracy-cost tradeoff. The advantage of BUDGETPRUNE is particularly large in (b). GREEDYMISER has lower accuracy in the high budget region compared to BUDGETPRUNE in (a) and significantly lower accuracy in (b). The gap between BUDGETPRUNE and other pruning methods is small in (a) but much larger in (b). This indicates large gains from globally encouraging feature sharing in the case of (b) compared to (a). In both datasets, BUDGETPRUNE successfully prunes away large number of features while maintaining high accuracy. For example in (a), using only 18 unique features on average instead of 40, we can get essentially the same accuracy as the original RF.

**Yahoo! Learning to Rank:[6]** This ranking dataset consists of 473134 web documents and 19944 queries. Each example in the dataset contains features of a query-document pair together with the

relevance rank of the document to the query. There are $141397/146769/184968$ examples in the training/validation/test sets. There are 519 features for each example; each feature is associated with an acquisition cost in the set $\{1, 5, 20, 50, 100, 150, 200\}$, which represents the units of CPU time required to extract the feature and is provided by a Yahoo! employee. The labels are binarized so that the document is either relevant or not relevant to the query. The task is to learn a model that takes a new query and its associated set of documents to produce an accurate ranking using as little feature cost as possible. As in [16], we use the Average Precision@5 as the performance metric, which gives a high reward for ranking the relevant documents on top. Our base RF consists of 140 trees using cost weighted entropy split criteria as in [16] and choosing from a random subset of 400 features at each split. As shown in (c) of Figure 2, BUDGETPRUNE achieves similar ranking accuracy as GREEDYMISER using only 30% of its cost.

**Scene15 [13]:** This scene recognition dataset contains 4485 images from 15 scene classes (labels). Following [22] we divide it into $1500/300/2685$ examples for training/validation/test sets. We use a diverse set of visual descriptors and object detectors from the Object Bank [14]. We treat each individual detector as an independent descriptor so we have a total of 184 visual descriptors. The acquisition costs of these visual descriptors range from 0.0374 to 9.2820. For each descriptor we train 15 one-vs-rest kernel SVMs and use the output (margins) as features. Once any feature corresponding to a visual descriptor is used for a test example, an acquisition cost of the visual descriptor is incurred and subsequent usage of features from the same group is free for the test example. Our base RF consists of 500 trees using entropy split criteria and choosing from a random subset of 20 features at each split. As shown in (d) of Figure 2, BUDGETPRUNE and GREEDYPRUNE significantly outperform other competing methods. BUDGETPRUNE has the same accuracy at the cost of 9 as at the full cost of 32. BUDGETPRUNE and GREEDYPRUNE perform similarly, indicating the greedy approach happen to solve the global optimization in this particular initial RF.

## 5.1 Discussion & Concluding Comments

We have empirically evaluated several resource constrained learning algorithms including BUDGET-PRUNE and its variations on benchmarked datasets here and in the Suppl. Material. We highlight key features of our approach below. (**i**) STATE-OF-THE-ART METHODS. Recent work has established that GREEDYMISER and BUDGETRF are among the state-of-the-art methods dominating a number of other methods [12, 21, 20] on these benchmarked datasets. GREEDYMISER requires building class-specific ensembles and tends to perform poorly and is increasingly difficult to tune in multi-class settings. RF, by its nature, can handle multi-class settings efficiently. On the other hand, as we described earlier, [12, 20, 21] are fundamentally "tree-growing" approaches, namely they are top-down methods acquiring features sequentially based on a surrogate utility value. This is a fundamentally combinatorial problem that is known to be NP hard [5, 21] and thus requires a number of relaxations and heuristics with no guarantees on performance. In contrast our pruning strategy is initialized to realize good performance (RF initialization) and we are able to globally optimize cost-accuracy objective. (**ii**) VARIATIONS ON PRUNING. By explicitly modeling feature costs, BUDGETPRUNE outperforms other pruning methods such as early stopping of BUDGETRF and CCP that do not consider costs. GREEDYPRUNE performs well validating our intuition (see Table. 1) that pruning sparsely occurring feature nodes utilized by large fraction of examples can improve test-time cost-accuracy tradeoff. Nevertheless, the BUDGETPRUNE outperforms GREEDYPRUNE, which is indicative of the fact that apart from obvious high-budget regimes, node-pruning must account for how removal of one node may have an adverse impact on another downstream one. (**iii**) SENSITIVITY TO IMPURITY, FEATURE COSTS, & OTHER INPUTS. We explore these issues in Suppl. Material. We experiment BUDGETPRUNE with different impurity functions such as entropy and Pairs [16] criteria. Pairs-impurity tends to build RFs with lower cost but also lower accuracy compared to entropy and so has poorer performance. We also explored how non-uniform costs can impact cost-accuracy tradeoff. An elegant approach has been suggested by [2], who propose an adversarial feature cost proportional to feature utility value. We find that BUDGETPRUNE is robust with such costs. Other RF parameters including number of trees and feature subset size at each split do impact cost-accuracy tradeoff in obvious ways with more trees and moderate feature subset size improving prediction accuracy while incurring higher cost.

**Acknowledgment:** We thank Dr Kilian Weinberger for helpful discussions and Dr David Castanon for the insights on the primal dual algorithm. This material is based upon work supported in part by NSF Grants CCF: 1320566, CNS: 1330008, CCF: 1527618, DHS 2013-ST-061-ED0001, ONR Grant 50202168 and US AF contract FA8650-14-C-1728.

## Footnotes

[1]When an example arrives at an internal node, the feature associated with the node is used to direct the example. If the feature has never been acquired for the example an acquisition cost is incurred. Otherwise, no acquisition cost is incurred as we assume that feature values are stored once computed.

[2]For time-sensitive cases such as web-search we parallelize the implementation by creating parallel jobs across all features and trees. We can then terminate jobs based on what features are returned.

[3]The nice result of totally unimodular constraints is due to our specific formulation. See Suppl. Material for an alternative formulation that does not have such a property.

[4] In contrast to traditional sparse feature selection, our algorithm allows adaptivity, meaning different examples use different subsets of features.

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
