[Reviews · NeurIPS 2016]

Reviewer 1

Summary

This paper target the problem of prediction on a budget: at prediction time there may not be sufficient time to compute all features based on raw data, thus this paper aims to adaptively acquire features as needed depending on the instance. It is a two-stage algorithm that first learns a random forest (RF), and then prunes the trees to allow faster prediction, i.e. by selectively pruning features sparsely used across trees. The latter is done by solving a 0-1 integer program with linear constraints, using linear program relaxation to obtain a polynomial-time algorithm. In itself this is not very novel, previous work [23, 18] introduced this method before. The main novelty lies in including feature costs, which is a harder problem to solve. The resulting LP can be too large to be practically solved by general-purpose LP solvers. The authors therefore introduce a primal-dual algorithm to decompose the problem into many subproblem that can be solved in parallel. The new algorithm is compared on 4 datasets with 3 other algorithms and a greedy baseline. There are several aspects of the evaluation that raise questions, but overal this seems like promising work.

Qualitative Assessment

Overall, this looks like a solid contribution. I do find the empirical evaluation not truly convincing for a number of reasons: * The evaluation procedure is described very vaguely. It seems that out-of-bag data is used to compute both test accuracy and feature cost, but this is never explicitly stated (nor are code or experiments shared). Is test accuracy the same as 1 minus the out of bag error? * The list of competing algorithms strikes me as odd. BudgetRF and CCP both prune trees using a heuristic designed to limit the depth of trees, not to reduce the number of features used. * Experiments are repeated 10 times and error bars are shown in the plots, but there is no statistical significance testing. In the same vein, it would be good to include more than 4 datasets. * There are no results reported on how much time the LP solver needs to prune the RF. * A smaller point: the authors remove ASTC [12], CSTC [21] and cost-weighted l-1 classifiers because they perform significantly worse in other papers. However, it is unclear whether those other papers use the same datasets or even a similar setup. Hence, it seems rash to disregard them entirely? Other remarks: * Page 2: "RF feature usage is superfluous, utilizing features with introduced randomness to increase diversity and generalization." This is simply not true? A RF selects data points and features randomly, but does not utilize 'features with introduced randomness'. Please state your point more clearly. * You claim that your bottom-up approach is better because 'top-down resource-constrained induction of decision trees is often intractable'. Can you state explicitly which methods you are referring to? It is [16], or others as well? * The paper layout seems wrong, with tables and figures added as wrap-around objects and very little whitespace. This should be fixed before the final submission.

Confidence in this Review

2-Confident (read it all; understood it all reasonably well)


Reviewer 2

Summary

This paper proposes a method for pruning a forest of trees taking into account feature costs. Pruning is formulated as a 0-1 integer program with linear constraints and it is shown that this program can be solved exactly through LP relaxation. Experiments are carried out on several datasets.

Qualitative Assessment

The idea of taking into account feature costs when pruning tree ensembles is original to the best of my knowledge. The main originality of the proposed approach is the fact that it adopts a bottom-up post-pruning strategy, while most existing approaches are top-down, acting during tree growing. While the authors present this feature as an advantage of their method, actually, I'm not convinced that adopting a bottom-up strategy is a good idea for addressing this problem. Since the algorithm indeed can not modify the existing tree structure (it can only prune it), it should be less efficient in terms of feature cost reduction than top-down methods that can have a direct impact on the features selected at tree nodes. For example, let us assume that two very important features in the dataset carry on the exact same information about the output (i.e, they are redundant). They can both be included in a forest if this latter is built without constraint and BudgetPrune will have no choice but to keep these two variables in the model, even though only one of them is actually necessary. The optimization problem is nicely formulated, in particular in its coupling of the feature usage variables. The proposed primal-dual optimization problem seems sound and very efficient as it scales linearly with respect to the number of trees (compared to the initial cubic LP relaxation). In the formulation of the (IP) problem, I'm surprised by the fact that the error term in the objective function is the average of the individual errors of the ensemble trees and not the error of the ensemble itself. I guess this is the main reason why the optimization problem can be solved efficiently by the primal-dual algorithm but I think that the authors should discuss and motivate this choice better in the paper. Another not so well-motivated design choice is the incorporation of the computational cost (average leaf depths) in the objective function. Given that the experiments only focus on assessing the average feature cost, is it necessary to include the computational cost? Doesn't it harm the average feature cost? Don't you need also to introduce an additional coefficient in the objective function to balance the feature acquisition costs and computational cost? Merely summing them seems risky without a proper normalization. Experiments seem to have been well conducted. They are extensive enough, although the number of datasets (4) could have been higher. They rather convincingly show the superiority of BudgetPrune over competitors (in Figure 2). Experiments in the supplementary material are also very useful to complement the experiments in the main paper. Some points however need clarification before the results can really be assessed: - The authors propose to use in-bag data to estimate error probability at each node and the validation data for the feature cost variables. It's not clear to me what it means. What is the in-bag data? Is it the learning sample itself or the bootstrap sample of each tree? - It's not clear if all methods make use of the validation data. For the methods that do not need a validation set, it would be more fair to train them on the whole training+validation data. - In the CCP method, what do the authors mean by the sentence "we perform CCP on individual trees to different levels to obtain various points on the accuracy-cost tradeoff curve"? This should be more precisely explained. - The GreedyPrune method is also unclear. What does it mean to remove all nodes corresponding to one feature? Do you prune the whole subtree below each split on this feature? If not, what do you do with the left-right successor nodes of this split? What does it mean to select the pruned RF that "has the lowest training error and average feature cost"? Do you sum these two scores? With a weight? - In the plots in Figure 2,how is the average feature cost measured on the x-axis? Also, how are different tradeoffs obtained for the different methods in Figure 2? With BudgetPrune, I guess that the parameter lambda is changed to obtain a curve but how is it done for the other methods? Is there only one parameter that can change feature cost? - All methods are evaluated only in terms of average feature cost and accuracy while the objective function also includes a term related to the computational costs (the average depth of the leaves reached by the examples). Why are the approaches not evaluated also according to this criterion? Is this term really crucial in the objective function? The paper is well-written and mostly clear (except for the missing details in the experiments). However, the fact that several experimental details are only present in the supplementary material makes reading section 5 not so easy. I think that some more details could have been given in the main paper. Minor comments: - There is no reference in the main text to the fact that the supplementary material contain additional (and necessary) details about the problems and settings of algorithms. A link should be given at the beginning of Section 5. - Page 1, I don't understand the meaning of footnote 2. - In the case of uniform cost, alternative competitors to BudgetPrune are feature selection methods that tries to find a small subset of features that are sufficient to predict the output. It would interesting to include feature selection methods in the comparison (for example using Random forests variable importances). - In Section 5.1: "This is a fundamentally combinatorial problem...and we are able to globally optimize cost-accuracy objective". "our proposed formulation possesses 1) elegant theoretical properties". The BudgetPrune algorithm indeed finds a global optimum but it needs to start from an existing random forests whose training is also a combinatorial problem solved in practice by using relaxations and heuristic with no guarantees of performance in the end. So, I'm not sure it's fair to say that BudgetPrune provides a better solution with stronger theoretical guarantees to the problem than top-down methods.

Confidence in this Review

2-Confident (read it all; understood it all reasonably well)


Reviewer 3

Summary

This paper addresses the challenge of prediction-time budget constraints and investigates the meta learning problem of balancing test time cost versus predictive accuracy. The authors propose a two-stage algorithm, which first trains a random forest classifier and then takes the classifier as input and attempts to jointly prune each tree in the forest to meet global resource constraints. The focus of the paper is on the second step. The authors define the feature cost of a test instance as the sum of costs of unique feature acquisitions across the forest. The pruning is formulated as a 0-1 integer linear program with constraints on feature reuse. The linear program relaxation makes it possible to solve the pruning problem in polynomial time. The proposed method is compared with 4 competing methods on 4 benchmark data sets. It is shown to outperform the competing methods trading off cost versus accuracy.

Qualitative Assessment

Technical quality: The description of the learner (random forests), the proposed method (BudgetPrune) seem to be complete and correct. The proposed method is well anchored to existing work. The experiments are simple and effective. The conclusions seem valid albeit the summarizing description of the contribution ("an algorithm scalable to large problems [with] superior empirical performance") is quite strong considering that the experiments only featured four datasets out of which none can be considered large with today's measures. Novelty/originality: In contrast to several recent methods, the proposed method uses a bottom-up approach that starts with good initialization (a random forest) and then prunes to realize optimal cost-accuracy tradeoff. One related paper also tries to solve the problem by pruning a generated random forests model but feature reuse is not considered. In other words, where most related work on test time cost reduction build custom classifiers or perform runtime pruning to approximate budget constraints, the proposed method is to perform post pruning instead. Although 0-1 integer linear programs have been proposed for random forest pruning earlier, such work did not consider feature cost. Impact/usefulness: The proposed method seems to be quite straight-forward to implement and, in the experiments, it outperforms the competing methods by achieving the best cost-accuracy trade-off on the 4 studied datasets. Methods for test time budget constraints will be increasingly important as many new applications involve very large datasets where obtaining feature values during prediction can be quite costly. Even though the authors do not explore the direction in this paper, the proposed approach (to use 0-1 integer linear programs for pruning models to achieve lower feature acquisition costs) could most likely be generalized to other model types and performance objectives. Clarity/presentation: In my opinion, the presentation is concise, effective, and convincing. The paper includes a few minor language mistakes but in general it is well written and correct. Additional comments/suggestions: - Is there a reason for formulating two similar sounding but clearly different measures for cost? (1 - the average number of unique features used by test examples, 2) the sum of acquisition costs of unique features) - Which of the two cost measures is the designated objective measure (expected feature cost)? - In line with the authors' argumentation, the overall accuracy of the forest does not seem to be greatly affected by the pruning but what happens in cases with high class imbalance where the "important class" is the extreme minority class? (Hypothetically, the most important factors for predicting such a minority class may seldom be evaluated for majority class instances) Minor remarks: l.117: "an subscript" (incorrect indefinite article) l.135: "Assume the acquisition cost [..] are given" (subject-verb number disagreement) Algorithm 1: "exmaple" (incorrect spelling)

Confidence in this Review

2-Confident (read it all; understood it all reasonably well)


Reviewer 4

Summary

This paper proposes to prune a random forest (RF) for resource-constrained prediction by optimizing expected feature cost and accuracy.

Qualitative Assessment

This paper proposes to prune a random forest (RF) for resource-constrained prediction by optimizing the expected feature cost and accuracy. A two-stage algorithm is proposed. In the first stage, a random forest (RF) of trees using existing impurity function is trained. In the second stage, the pruning problem is formulated as a 0-1 integer linear program that incorporates feature-reuse constraints, whose linear program relaxation could yield the optimal solution to the integer program resulting in a polynomial time primal-dual algorithm. The problem is a worthy problem and the solutions are reasonable. As a premise, I must say that I am not an expert in the resource-constrained learning field, so I reviewed the paper with the general interest of a researcher in machine learning. My overall impression is positive, the idea seems interesting from a practical point of view. As I am not so familiar with the resource-constrained learning, I cannot judge whether formulating the problem as IP in this paper is new or the following transformation of IP constraints to network matrix is new. If this is the case, then I think the solution is nice. The presentation is overall clear and I could follow the presented ideas, with some exceptions: 1) the formatting like places of Table 1 and Figure 2 is bad; 2) the term "network matrix" is introduced and used in paragraph "totally unimodular constraints" and Lemma 3.1 without explaining the benefits of turning the constraints into network matrix, which is confusing. There are several concerns of mine on the experiment section: 1) while the paper gives detailed descriptions and training/testing setting on the Yahoo and Scene15 dataset, such descriptions of the MiniBooNE and Forest Covertype datasets are lacked; 2) while the paper state the setting such as the number of trees and their split criteria for the proposed approach, such implementation detailed information for the compared baselines are lacked; 3) I found that the implemented setting of the proposed approach including the size of trees and splitting criteria on the four datasets are different. If such implementations are selected to get superior performance over baselines, such comparison is biased and unconvincing.

Confidence in this Review

2-Confident (read it all; understood it all reasonably well)


Reviewer 5

Summary

In the paper, the authors try to decrease the feature acquisition cost and computational cost by pruning a random forest. The authors are able to formulate the pruning problem into an integer programming with considering the error rate, feature acquisition cost and computational cost. Then a primal-dual algorithm is proposed to solve the integer programming problem. The experimental results of test accuracy vs feature cost show that the pruning algorithm in this paper outperforms others.

Qualitative Assessment

1. In the integer programming formulated in the paper (page 4), w_{k,i} is treated as a variable that the objective minimizes over. However, from my understanding, when the examples are known, w_{k,i} is a constant. And in the feature acquisition cost in the objective should use w_{w,i}^{(t)} instead of w_{k,i} in order to measure the acquisition cost of the random forest after pruning. Similarly, the computational cost in the objective should use z_h^{(t)} instead of z_h. 2. The point of doing pruning is to reduce the feature cost during the prediction while preserving the accuracy. In the experiments section, it would be nice to include some results of computational time of the pruning algorithm vs the number of examples, and the computational time the pruning saves in the prediction vs the number of testing samples. This can give the readers an idea about when doing a pruning can be beneficial.

Confidence in this Review

2-Confident (read it all; understood it all reasonably well)


Reviewer 6

Summary

In their paper 'Pruning Random Forests for Prediction on a Budget', the authors present a new pruning mechanism to reduce the prediction time without sacrificing accuracy. This is a relative problem when features are associated with a cost at prediction time. In the introduction the approach is properly motivated and set into context with the existing literature. Section 2 and 3 lay out th method in great detail. An integer program (IP) is formulated encoding any valid pruning of the forest. By showing the total unimodularity of the constraints, finding the optimal pruning becomes feasible using a primal-dual algorithm. In the experimental section, this method is compared to a simple greedy method and other state of the art approaches for constrained predictions. It should be noted that only the authors' BudgetPrune and the GreedyPrune actively take the feature cost into account. The results on a few datasets show that the algorithm performs well in different circumstances. The supplementary material provides the proof for Theorem 3.2, and additional experiments investigating different random forest variants as the initialization for BudgetPrune.

Qualitative Assessment

The paper is seemingly the first attempt to prune trees with feature costs in mind, which will become more and more relevant in times of big data. The presentation is very good and the technical level quite high, if one is not familiar with random forests and integer programming. Remarks/Questions: I found Table 1 and its caption troubling. It says that before pruning, the feature is only used a few times (1-7) 'paying a significant cost for its acquisition'. The pruning drastically increases the fraction of examples that don't have to compute the feature at all leading to a drop in the cost, but the majority of the examples still use the feature at most 7 times if they do. In fact, the ratio of examples reusing this particular feature many times goes down, which partially 'contradicts' motivation (iii). I totally buy the motivation and the methodology, so I was puzzled by that. Could 7 be already too large as a reference for 'many times'? The illustrative example around Figure 1 is very helpful, but I struggled with one maybe crucial detail: Why do the z_i not sum to one in every row? Even more surprising to me was the fact that z_1 is always equal to 1. Doesn't that mean the root is a leaf and any other node below is pruned? In the discussion of the results for Scene15, it is stated that GREEDYPRUNE performs similarly because the 'greedy approach happen(s) to solve the global optimization of this particular RF'. Do you have any insight why? Maybe only a few features are important, and both BudgetPrune and GreedyPrune successfully identify them easily (because they are both bottom-up approaches? In general, it would be really helpful to see the initial RFs point in the plots as a reference. Minor remarks: Sometimes, the authors use citations as nouns, which is not following the style guide (section 4.1, Citations within the text) . See, for example, line 72 or 84. Section 2 is really short. Is there now way to incorporate that into 3? I know that space it tight, but the text formatting around figure 2 is really hard to read. The primal-dual discussion could go into the supplementary material.

Confidence in this Review

2-Confident (read it all; understood it all reasonably well)